

# The tale of springs and streams: how different aquatic ecosystems impacted the mtDNA population structure of two riffle beetles in the Western Carpathians

Jana Bozáňová[1,2], Zuzana Čiamporová Zaťovičová[2], Fedor Čiampor Jr[2], Tomasz Mamos[3,4] and Michał Grabowski[3]

[1] Department of Ecology, Faculty of Natural Sciences, Comenius University in Bratislava, Bratislava, Slovak Republic
[2] ZooLab, Plant Science and Biodiversity Centre, Slovak Academy of Sciences, Bratislava, Slovak Republic
[3] Department of Invertebrate Zoology and Hydrobiology, Faculty of Biology & Environmental Protection, University of Łódź, Łódź, Poland
[4] Zoological Institute, University of Basel, Basel, Switzerland

Corresponding author
Zuzana Čiamporová Zaťovičová,
zuzana.zatovicova@savba.sk

## ABSTRACT

The Western Carpathians are a particularly interesting part of the Carpathian Arc. According to recent molecular data upon aquatic and terrestrial taxa, this mountain area is an important biodiversity hotspot of Europe. Moreover, the W Carpathians include rich systems of karst springs inhabited by specific fauna, where molecular diversity and phylogeographic patterns are yet to be fully explored. Our study aims to compare population genetic structure and molecular diversity of two related and commonly co-occurring riffle beetles, *Elmis aenea* (PWJ Müller, 1806) and *Limnius perrisi* (Dufour, 1843) in the springs and streams of the W Carpathians using the mitochondrial DNA barcoding fragment of the cytochrome c oxidase subunit I gene (COI). The relatively stable thermal and chemical conditions of springs throughout unfavourable climatic settings make these highly specific lotic systems potentially ideal for a long-term survival of some aquatic biota. Populations of both elmid species were relatively homogeneous genetically, with a single dominant haplotype. However, we revealed that *E. aenea* significantly dominated in the springs, while *L. perrisi* preferred streams. Relative isolation of the springs and their stable conditions were reflected in significantly higher molecular diversity of the *E. aenea* population in comparison to *L. perrisi*. The results of Bayesian Skyline Plot analysis also indicated the exceptional position of springs regarding maintaining the population size of *E. aenea*. On the other hand, it seems that streams in the W Carpathians provide more effective dispersal channels for *L. perrisi*, whose population expanded much earlier compared to *E. aenea*. Present study points out that different demographic histories of these two closely related elmid species are manifested by their different habitat preference and molecular diversity.

## INTRODUCTION

Prolonged isolation of populations influences their genetic diversity and can be considered as the main force shaping genetic structure of aquatic species in Europe (*Bálint et al., 2011*; *Alp et al., 2012*; *Theissinger et al., 2012*). Isolated populations of aquatic invertebrates may be found within geomorphological units/subunits of many mountain areas (*Engelhardt, Haase & Pauls, 2011*; *Davis et al., 2013*; *Mamos et al., 2016*; *Čiamporová-Zaťovičová & Čiampor Jr, 2017*; *Šípošová, Čiamporová Zaťovičová & Čiampor Jr, 2017*; *Copilaş-Ciocianu et al., 2018*).

The W Carpathians are considered a biodiversity hotspot for a wide range of aquatic and terrestrial taxa (*Neumann et al., 2005*; *Kotlík et al., 2006*; *Theissinger et al., 2012*; *Vörös et al., 2016*; *Copilaş-Ciocianu et al., 2017*; *Juřičková et al., 2017*). However, the biodiversity of the W Carpathians is still underexplored, especially in terms of genetic diversity and population structure of aquatic species. In this context, studies upon the phylogeography of aquatic biota should be more focused on springs, or more generally, headwaters that are now heavily understudied compared to other aquatic biotopes.

Springs support unique macroinvertebrate communities that are found nowhere else in a catchment (*Lewin et al., 2015*). They are characterized by chemical, physical, and trophic constancy over several geological periods (*Minshall & Winger, 1968*; *Odum, 1971*; *Butler & Hobbs, 1982*; *Cushing & Wolf, 1984*; *Glazier & Gooch, 1987*; *Pringle et al., 1988*; *Gooch & Glazier, 1991*; *Orendt, 2000*; *Wood et al., 2005*; *Meyer et al., 2007*), which in turn provided a stable environment for aquatic invertebrates during adverse climatic conditions (*Malicky, 2006*; *Ujvárosi et al., 2010*). Springs function as ecotones between the surface and underground waters, which makes them an ecologically significant habitat (*Gibert, 1991*).

Accordingly, the main objective of our study is to compare genetic population structure and diversity patterns of two aquatic beetle species of Elmidae family, *Elmis aenea* (PWJ Müller, 1806) and *Limnius perrisi* (Dufour, 1843) in springs and streams of the W Carpathians. Said species are relatively closely related and commonly co-occur, yet in terms of population genetics represent a generally understudied family of freshwater beetles. Limited dispersal abilities, high habitat specificity, and more or less fragmented distribution make Elmidae an ideal taxon for studying genetic diversification through many geographic regions. Both studied species are rheophilic, oligo-stenotherm, and typical inhabitants of epirhithral streams at higher altitudes (*Moog & Jäch, 1995*; *García-Criado, Fernández-Aláez & Fernández-Aláez, 1999*). They are relatively widespread, which guarantees the detection of possible gene flow among geomorphological units/subunits of W Carpathians.

Our study aims to answer the following questions: (a) Are the spring subpopulations genetically more variable when compared to subpopulations in the streams? (b) Does genetic structuring of populations reflect population size change? (c) Are there interspecific differences in the population genetic structure among these related beetle species?

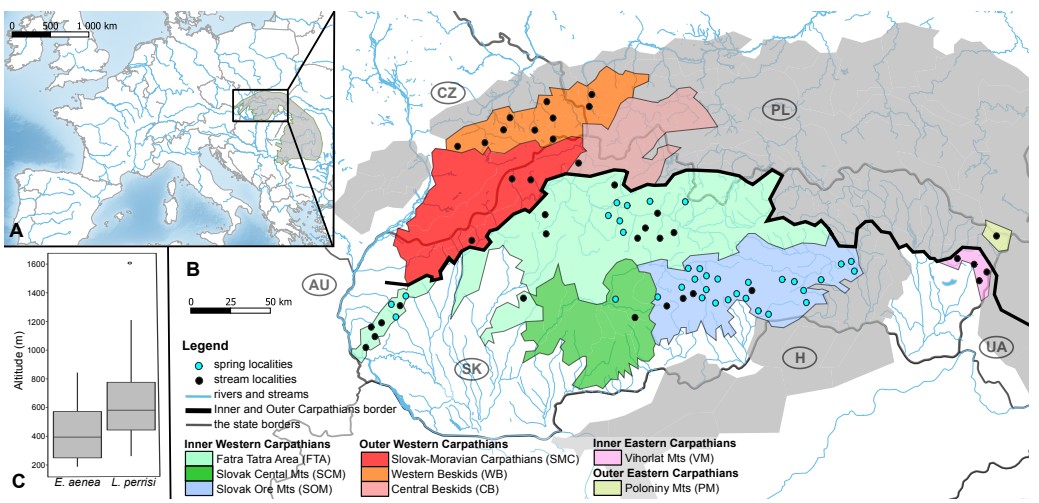

**Figure 1** **Maps of the studied area and sampling sites.** (A) Map of the studied area within the Carpathian Arc and (B) the 73 sampling sites (36 springs and 37 streams) divided into eight geomorphological units represented by different fill colors. (C) The altitude range of both elmid species. The boxplots show the distribution of the altitude above sea level for *Elmis aenea* and *Limnius perrisi*. The boxes represent the interquartile distances (IQD), while the centre lines through each box show the medians. The dot indicates outliers and the whiskers extend to the extreme values of the data, calculated as $\pm 1.5 \times$ IQD from the median. ANOVA analysis supported the dependence of species presence on altitude ($P < 0.05$). Abbreviations: Slovakia (SK), Hungary (H), Ukraine (UA), Poland (PL), Czech Republic (CZ) and Austria (AU).

## MATERIALS & METHODS

### Study area

The Carpathians form an arc of mountains stretching across Central and Eastern Europe, with its main geomorphological units being the Western and Southeastern Carpathians (*Kondracki, 1989*). For this study, we focused on the W Carpathians representing the northernmost segment of the Alpine-Carpathian mountain chain (Fig. 1A). The W Carpathians reach medium altitudes (ranging from 500 to 1,300 m a.s.l.), only a few of their ranges exceed 1,500 m a.s.l.; geologically the mountain system is characteristic by interactions of rock folding and horizontal shifts (*Bielik, 1999*). Differences in altitude and in the geomorphological relief determine the precipitation in the area. In general, W Carpathian rivers have a rain-snow regime with floods in spring and summer.

Investigated localities (36 springs and 37 streams) are situated mainly on the territory of the Slovak Republic, partially in the Czech Republic and Poland; in the geomorphological units/subunits of the Inner and Outer Western Carpathians. The exceptions are Vihorlat Mts (VM) being part of the Inner Eastern Carpathians and Poloniny Mts (PM) belonging to the Outer Eastern Carpathians (Fig. 1B, Table S1).

### Sampling and morphological identification

A qualitative sampling of benthic invertebrates took place in 2016 and 2017. The sampling was performed in the framework of a broader research, which was permitted by The

District Office, Department of Environmental Care Trenčín (Slovakia) No: OU-TN-OSZP1-2015/001937-12/Du. A sampling of macrozoobenthos was carried out by a multi-habitat kick-sampling technique (*Frost, 1971*) using a hydrobiological hand-net with a mesh size of 0.5 mm. Organic material was fixed in 96% ethanol directly in the field. In the laboratory, the invertebrates were picked off, sorted into higher taxonomic groups using stereomicroscope, prefixed with absolute ethanol and stored in a freezer at −25 °C. Elmidae beetles selected for molecular analysis were morphologically identified using the available determination keys (*Wiȩźlak, 1986*; *Jäch, 1992*).

## DNA extraction and PCR amplification

Total DNA was extracted from the legs or abdominal tissue of 560 individuals (297 individuals of *E. aenea*; 263 individuals of *L. perrisi*) using the Chelex protocol (*Casquet, Thebaud & Gillespie, 2012*), followed by PCR amplification of ca. 650 bp-long barcoding fragment of the mitochondrial cytochrome c oxidase subunit I (COI) using the primer pair LCO1490 and HCO2198 (*Folmer et al., 1994*). The PCR was performed in a total volume of 25 μl containing 5 μl of 5× DreamTaq$^{TM}$ Buffer, 1.5 μl of Mg$^{+2}$ (25 mM), 0.5 μl of each primer (concentration 5 mM), 0.5 μl of dNTP Mix (20 mM), 0.125 μl (0.625 U) DreamTaq *TM* DNA Polymerase, 11.875 μl ultra-pure H$_2$O and 5 μl of DNA template. The PCR cycling consisted of a 2-min initial denaturation at 94 °C, followed by 40 cycles of 94 °C (40 s) denaturation, 46 °C (40 s) annealing and 72 °C (1 min) extension and termination at 72 °C (10 min) for a final extension. A 4 μl aliquot of the PCR products were visualized by GoldView (Solarbio) in electrophoresis on a 1% agarose gel and GelLogic imaging equipment to check PCR product quality and length. The PCR products were purified with Exo-FastAP Thermo Scientific and were sent for sequencing to Macrogen Europe Inc., Amsterdam.

## Data analyses

To determine whether the different habitat preference (springs, streams) between the studied species is statistically significant, we used Fisher's exact tests using the fisher.test function in R v4.0.2 (http://www.r-project.org). It was performed for testing the independence of rows (species: *E. aenea*, *L. perrisi*) and columns (springs, streams) in a 2 × 2 contingency table. Odds ratio and *p*-value were computed. *P*-values <0.05 were considered statistically significant. We used analysis of variance (ANOVA) to test significance of the influence of altitude on the presence of species in different habitats (springs, streams). ANOVA test was performed in R v4.0.2 (http://www.r-project.org). All R analysis were carried out using RStudio (*RStudio Team, 2020*). The altitude range of *E. aenea* and *L. perrisi* is shown with boxplots.

The obtained sequences were edited using SEQUENCER v5.1 software and aligned using the MUSCLE algorithm (*Edgar, 2004*) in MEGA v7 (*Kumar, Stecher & Tamura, 2016*). In total, our study included 315 sequences of *E. aenea* species, of which 276 were from the Western Carpathians and 269 sequences of *L. perrisi*, of which 245 were from the Western Carpathians. We used 39 sequences of *E. aenea* (12 - Romania, 9 - Bulgaria, 15 - Germany, 2 - Finland, 1 - France) and 24 of *L. perrisi* (16 - Romania, 2 - Bulgaria, 6 -

Germany) outside of W Carpathians for haplotype networks. Sequences from Germany, Finland and France were downloaded from BOLD (http://www.boldsystems.org) and are included in datasets DS-SKLIMPER (DOI: dx.doi.org/10.5883/DS-SKLIMPER) and DS-SKELMAEN (DOI: dx.doi.org/10.5883/ DS-SKELMAEN). For the purposes of the paper, individuals (sequences) from each locality of W Carpathians are defined as subpopulation regardless of whether the locality is a spring or stream (Table S1).

The haplotype data files and the diversity indices were generated in DnaSP v5.10 (*Librado & Rozas, 2009*). We also calculated haplotype diversity (H), nucleotide diversity ($\pi$), number of polymorphic sites (S) and average number of nucleotide differences (K) per subpopulation of both species using DnaSP v5.10. Subsequently, a statistical comparison of molecular genetic indices between species, based on *p*-values, was computed with the Wilcoxon signed rank test for paired data in R v4.0.2 (http://www.r-project.org). The results are presented by boxplots with *p*-values.

Haplotype networks were reconstructed using the median-joining method (MJN) in PopART v1.7 (*Leigh & Bryant, 2015*). The networks include some sequences outside the W Carpathians to explain the phylogenetic relationships and haplotype distribution of *E. aenea* and *L. perrisi* in the broader context of the investigated localities.

The population structure of both species was characterized by the analysis of molecular variance (AMOVA) and fixation indices ($F_{ST}$) using Arlequin v3.5 (*Excoffier & Lischer, 2010*). The AMOVA was used to estimate whether the observed genetic diversity may be attributed to the geographical partitioning of elmid beetle populations in three levels: among geomorphological subunits, among subpopulations within subunits and within subpopulations. For the consistency of the study, we also performed AMOVA for both species based on the partitioning of the data according to river basins. The results and map showing the river basins are included in the supplementary materials.

$F_{ST}$ index is a measure of the genetic differentiation among subpopulations of individual localities by haplotype frequencies. 265 *E. aenea* sequences (42 localities–29 springs, 13 streams) and 136 *L. perrisi* sequences (36 localities–5 springs, 31 streams) were included to calculate the $F_{ST}$ index. Localities with 1 sequence were excluded from the calculation. To test the significance of covariance components and fixation indices, 1,000 permutations were performed.

To test if spatial distance is structuring the molecular diversity we run two types of isolation by distance tests: Mantel test (*Mantel, 1967*) and general spatial autocorrelation test using the program Alleles in Space (*Miller, 2005*). Both tests analyse correlation between spatial and molecular distance, to assess the significance tests were run with 1,000 permutations.

Further, the demographic and spatial dynamics of studied beetle populations were examined by the mismatch distribution analysis in Arlequin v3.5. The recent demographic expansion in both species was tested with Tajima's D (*Tajima, 1989*), Fu's Fs (*Fu, 1997*) and Fu and Li's D (*Fu & Li, 1993*) tests of selective neutrality and population stability, performed in DnaSP. The significance of these tests was assessed with 10,000 permutations.

The fluctuations of demography of *E. aenea* and *L. perrisi* in the W Carpathians over time were identified with the extended Bayesian Skyline Plot (eBSP) in BEAST v2.6.2

software package (*Bouckaert et al., 2019*). The strict molecular clock was calibrated with the standard mitochondrial rate for arthropod COI equal to 0.0115 substitutions/site/Myr (*Brower, 1994*). The models of molecular evolution were set up through bModelTest (*Bouckaert & Drummond, 2017*). For comparison, two runs of Monte Carlo Markov Chains (MCMC) were performed for each species, each 40 million iterations long and sampled every 10,000 iterations for eBSP log. The runs were examined in Tracer v1.7 (*Rambaut et al., 2018*) and all the parameters reached the effective sampling size (ESS) above 200. After removal of 10% burn-in, the eBSP plots were produced using R v4.0.2 software (http://www.r-project.org). Both plots for each species were identical therefore only one is presented.

All analysed sequences with GenBank accession numbers are available within two BOLD datasets: DS-SKLIMPER for *L. perrisi* (DOI: dx.doi.org/10.5883/DS-SKLIMPER) and DS-SKELMAEN for *E. aenea* (DOI: dx.doi.org/10.5883/DS-SKELMAEN).

# RESULTS

The distribution of *E. aenea* and *L. perrisi* suggests statistically significant different habitat preferences between these species in the W Carpathians ($p < 0.0001$; Fisher's exact test). *E. aenea* has a rather wide distribution in karst springs (31 sites), while it is less widespread in streams (16 sites). On the contrary, *L. perrisi* was found only in eight springs, but in 30 streams. *L. perrisi* was also found in four streams of VM (Inner Eastern Carpathians) and in one stream of PM (Outer Eastern Carpathians), while *E. aenea* was not recorded in these geomorphological subunits. Both species co-occurred only in three springs and 14 streams from the total of 73 sites sampled in the W Carpathians (Table S1). There was also a significant effect of altitude on the presence of both species registered, but the dependence between altitude and habitat type (spring and stream) has not been demonstrated. The altitude range of both elmid species is shown by boxplots in Fig. 1C.

The haplotype distribution within investigated area shows that local subpopulations of the two elmid species are dominated, each, by the same widespread haplotype (Figs. 2A, 3A). Haplotype networks (Figs. 2B, 3B) also showed the similar haplotype pattern i.e., star-like topology with a central most-frequent haplotype. However, statistical comparisons of molecular genetic indices: haplotype diversity (H), nucleotide diversity ($\pi$), number of polymorphic sites (S), and average number of nucleotide differences (K) showed significant differences between the studied species. The population of *E. aenea* was significantly more diverse than *L. perrisi* ($P < 0.05$, Wilcoxon signed-rank test, Fig. 4).

The W Carpathian population of *E. aenea* shares haplotypes with locations in Romania, Bulgaria, Finland, Germany and France. 13 COI haplotypes of *E. aenea* were identified within 276 individuals collected from 47 localities in the W Carpathians (Fig. 2A). The haplotype diversity was 0.336. Considerable genetic homogeneity of the *E. aenea* population in the W Carpathians resulted from the wide distribution of the dominant haplotype Ea1. The haplotype map (Fig. 2B) revealed that the majority of haplotypes present in southern part of the Carpathians Arc (Romania) and the haplotypes of the Balkan region (Bulgaria) were not recorded from the W Carpathians. The exception was Ea14 shared between one
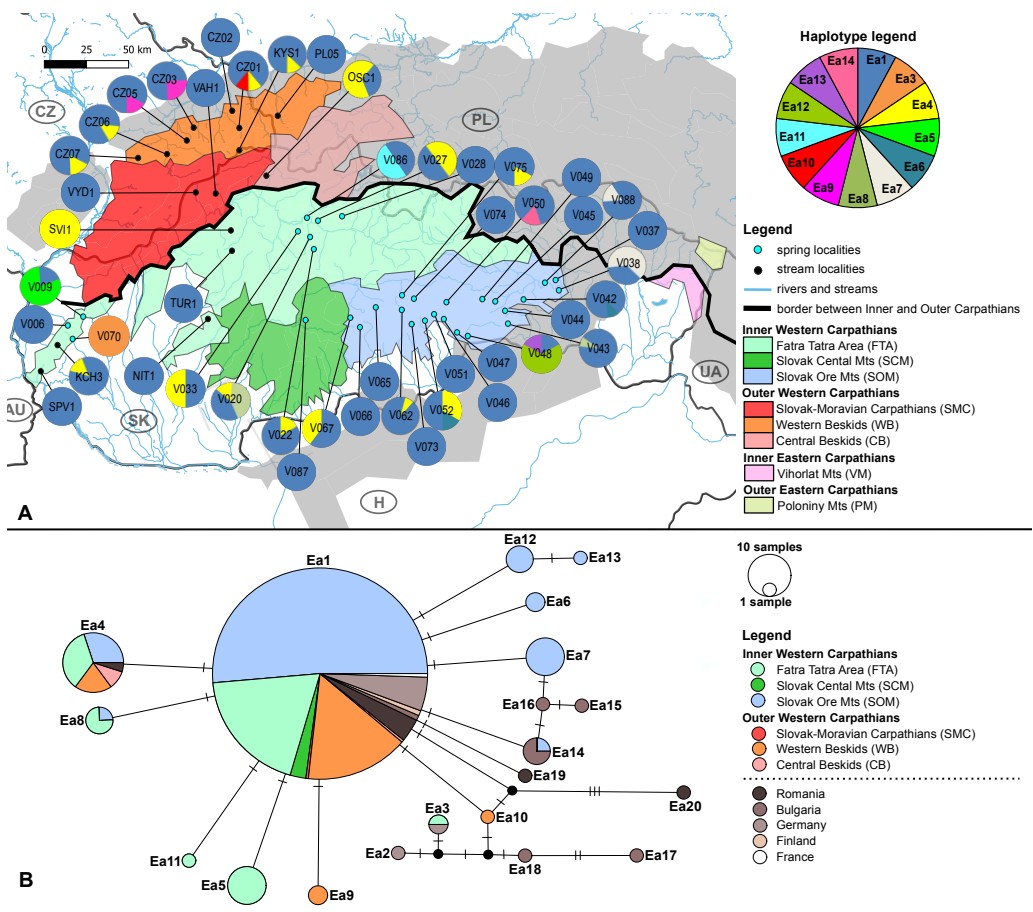

**Figure 2** *Elmis aenea* **sampling sites with mtDNA (COI) haplotype distribution and haplotype network.** (A) Investigated springs (31) and streams (16) with 13 mtDNA haplotypes distribution. Haplotype colors follow the Haplotype legend. Geomorphological units are represented by different fill color according to the legend. (B) Median-Joining network showing the relationships among haplotypes Ea1 - Ea20 (including available haplotypes outside W Carpathians). Sequences from Romania (12 sequences), Bulgaria (nine sequences), Finland (two sequences), Germany (15 sequences) and France (one sequence) are used for suggesting possible phylogenetic relationships and haplotype distribution of the W Carpathian haplotypes in the broader context. Circle fill colors follow the legend. Mutational steps are indicated with bars, small black dots represent undetected haplotypes.

stream in Bulgaria and one spring (V050) in Slovakia (SOM). Individuals from Germany shared the haplotype Ea3 with a single locality in the geomorphological unit FTA (V070). In addition to dominant haplotype Ea1, another five haplotypes were found in FTA and seven in SOM. Haplotypes Ea5 and Ea11 were private for W Carpathians and each occurred in one spring of FTA (V009, V086). The private haplotypes of SOM included Ea7, Ea12 and Ea13, while all of them were located in the springs of SOM1 subunit (V038, V048). Besides that, one spring of SOM1 (V043) shared haplotype Ea8 with the spring of the geomorphological unit FTA (V020). In geomorphological unit WB, four haplotypes were found, while Ea9 was found only at two localities (CZ03, CZ05). On the contrary, the haplotype Ea4 was common in the SOM, FTA, CB and in Romania (Figs. 2A, 2B).

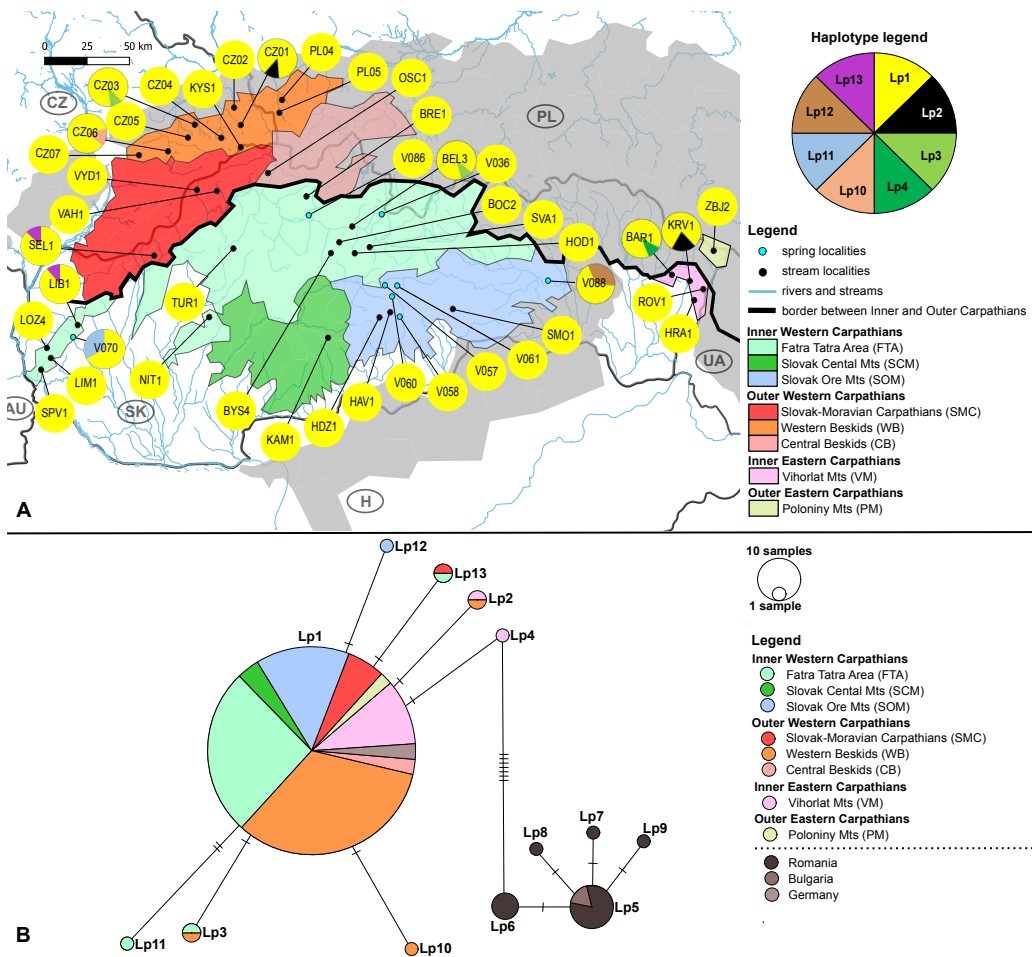

**Figure 3** *Limnius perrisi* **sampling sites with mtDNA (COI) haplotype distribution and haplotype network.** (A) Investigated springs (eight) and streams (35) with 7 mtDNA haplotypes distribution. Haplotype colors follow the Haplotype legend. Geomorphological units are represented by different fill colors according to the legend. (B) Median-Joining network showing the relationships among haplotypes Lp1 - Lp13. Sequences from Romania (16 sequences), Bulgaria (two sequences) and Germany (six sequences) are used for suggesting the possible phylogenetic relationships and haplotype distribution of the W Carpathian haplotypes in the broader context. Circle fill colors follow the legend. Mutational steps are indicated with bars, small black dots represent undetected haplotypes.

The W Carpathians population of *L. perrisi* was genetically more homogeneous. Eight haplotypes with a haplotype diversity of 0.007 were found at 43 localities (245 sequences, Fig. 3A). A group of five haplotypes (Lp5, Lp6, Lp7, Lp8, Lp9) recorded in Romania and Bulgaria was highly divergent from the group found in the W Carpathians (Fig. 3B). Haplotype Lp1 dominated in all geomorphological units of the W Carpathians; all German sequences also belonged to this haplotype. The presence of private haplotypes was lower compared to *E. aenea*: Lp14 from the one spring of SOM (V088), Lp10 in a stream of WB (CZ06) and Lp11 from one spring of geomorphological unit FTA (V070). Besides that, two more haplotypes (Lp3, Lp13) were present in the FTA. Lp13 was shared with the locality of the different geomorphological unit SMC (SEL1). Lp3 was also present in unit WB, in

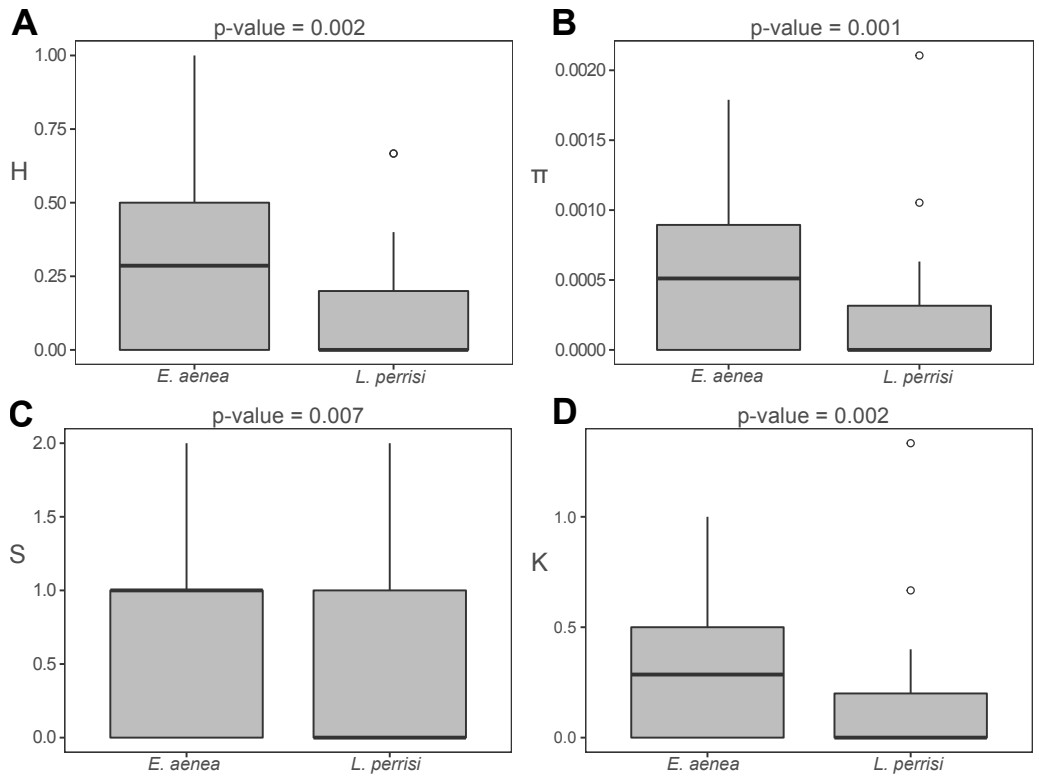

**Figure 4** **Comparison of molecular diversity indices between *Elmis aenea* and *Limnius perrisi* populations in Western Carpathians.** Box plots show (A) the haplotype diversity, (B) nucleotide diversity, (C) number of polymorphic sites and (D) average number of nucleotide differences. The statistical significance was computed with the Wilcoxon signed rank test for paired data (*p*-value: above each box plot).

addition to FTA (BEL3). The haplotype Lp2 was detected in a stream (CZ01) of WB and occurred also in one stream (KRV1) located in VM (Figs. 3A, 3B).

The AMOVA showed that most of the observed molecular variance in the W Carpathian populations of both elmid species is generated predominantly within subpopulations (single localities). However, in *E. aenea*, the molecular variation among subpopulations within geomorphological subunits is more than twice (34.99%) compared to *L. perrisi* (12.86%) (Table 1). The same results were provided by the AMOVA according to the river basins, where there is also almost no variation associated with the above sub-population (locality) level (Fig. S1, Table S2).

The $F_{ST}$ values indicate different levels of genetic differentiation between the *E. aenea* localities 0–0.8 (Fig. 5A) compared to *L. perrisi* with 0–0.38 (Fig. 5B). The $F_{ST}$ values of *E. aenea* suggest that springs V009 (FTA1 - Little Carpathians), V038, and V048 (SOM1 - Slovak Karst) have relatively high pairwise differences in allele frequency, but some level of genetic connectivity cannot be refused. Pairwise comparisons of differences in the frequency of alleles that include mentioned springs are largely significant ($P < 0.05$). On the other hand, none of *L. perrisi* $F_{ST}$ values are statistically significant.

**Table 1** Analysis of molecular variance (AMOVA) calculated from 273 COI mtDNA sequences of *Elmis aenea* and 245 COI mtDNA sequences of *Limnius perrisi* from studied springs and streams in the W Carpathians. Subunits = geomorphological subunits. The subpopulation is defined as individuals of one sampling site, Table S1.

| Source of variation | df[a] | SS[b] | Variance components | % of variation | F value | *p*-value |
|---|---|---|---|---|---|---|
| *E. aenea* | | | | | | |
| Among subunits | 10 | 5.704 | 0.00206 | 1.02 | $F_{CT} = 0.010$ | > 0.352 |
| Among subpopulations within subunits | 36 | 19.768 | 0.07080 | 34.99 | $F_{SC} = 0.353$ | > 0.000 |
| Within subpopulations | 230 | 29.784 | 0.12950 | 64 | $F_{ST} = 0.360$ | < 0.000 |
| *L. perrisi* | | | | | | |
| **Source of variation** | **df[a]** | **SS[b]** | **Variance components** | **% of variation** | **F value** | **p-value** |
| Among subunits | 12 | 0.657 | −0.00148 | −3.16 | $F_{CT} = 0.010$ | > 0.335 |
| Among subpopulations within subunits | 29 | 2.145 | 0.00602 | 12.86 | $F_{SC} = 0.125$ | > 0.097 |
| Within subpopulations | 192 | 8.117 | 0.04227 | 90.3 | $F_{ST} = 0.097$ | < 0.074 |

**Notes.**
[a]df, Degree of freedom.
[b]SS, Sum of squares.

Tests of isolation by distance between springs of both species revealed a positive correlation (Mantel test: *E. aenea* - $r = 0.313$, $P = 0.000$; *L. perrisi* - $r = 0.4122$, $P = 0.039$). Although, only marginally positive but statistically significant correlations suggest a slight structuring effect of the geographical distance among springs in both species. Additionally in case of *E. aenea*, the spatial autocorrelation was also significant ($P = 0.0009$, Fig. S2), for *L. perrisi* it was impossible to calculate it due to lack of data. The genetic distance of *E. aenea* and *L. perrisi* in streams was not significantly correlated with the geographic distance (Mantel test: $r = -0.071$, $P = 0.153$; $r = 0.0574$, $P = 0.115$).

Both species were characterized by the statistically significant, negative Fu's Fs, Tajima's D and Fu and Li's D neutrality test values (Table 2). This indicates a recent change in population size of both species. The mismatch distribution analysis suggested a population expansion event for both species, which was indicated by the unimodal shape of the mismatch distribution plot, a small SSD value, and a non-significant *p*-value (Fig. 6). The eBSP showed a signal of population growth in both species, although the time and character was different. The W Carpathians population of *E. aenea* (Fig. 7A) started to expand demographically roughly ca. 3,000–2,500 years ago, whereas the population of *L. perrisi* expanded relatively sharply around 8,000 years ago (Fig. 7B).

## DISCUSSION

This study was focused on the two oligo-stenotherm riffle beetles, *Elmis aenea* and *Limnius perrisi* (Elmidae). The *E. aenea* occurred predominantly in karst springs and was rarely found in streams of the W Carpathians, while the distributional pattern of *L. perrisi* was opposite. This contradicts the previous claims about their common occurrence and similar biotope preference (*Moog & Jäch, 1995*; *García-Criado, Fernández-Aláez & Fernández-Aláez, 1999*). Differences in distribution probably can be also explained by altitude, flow type or different ecological demands (*Illies & Botosaneanu, 1963*). However, according to several studies, *E. aenea* is more sensitive to harsher conditions resulting from

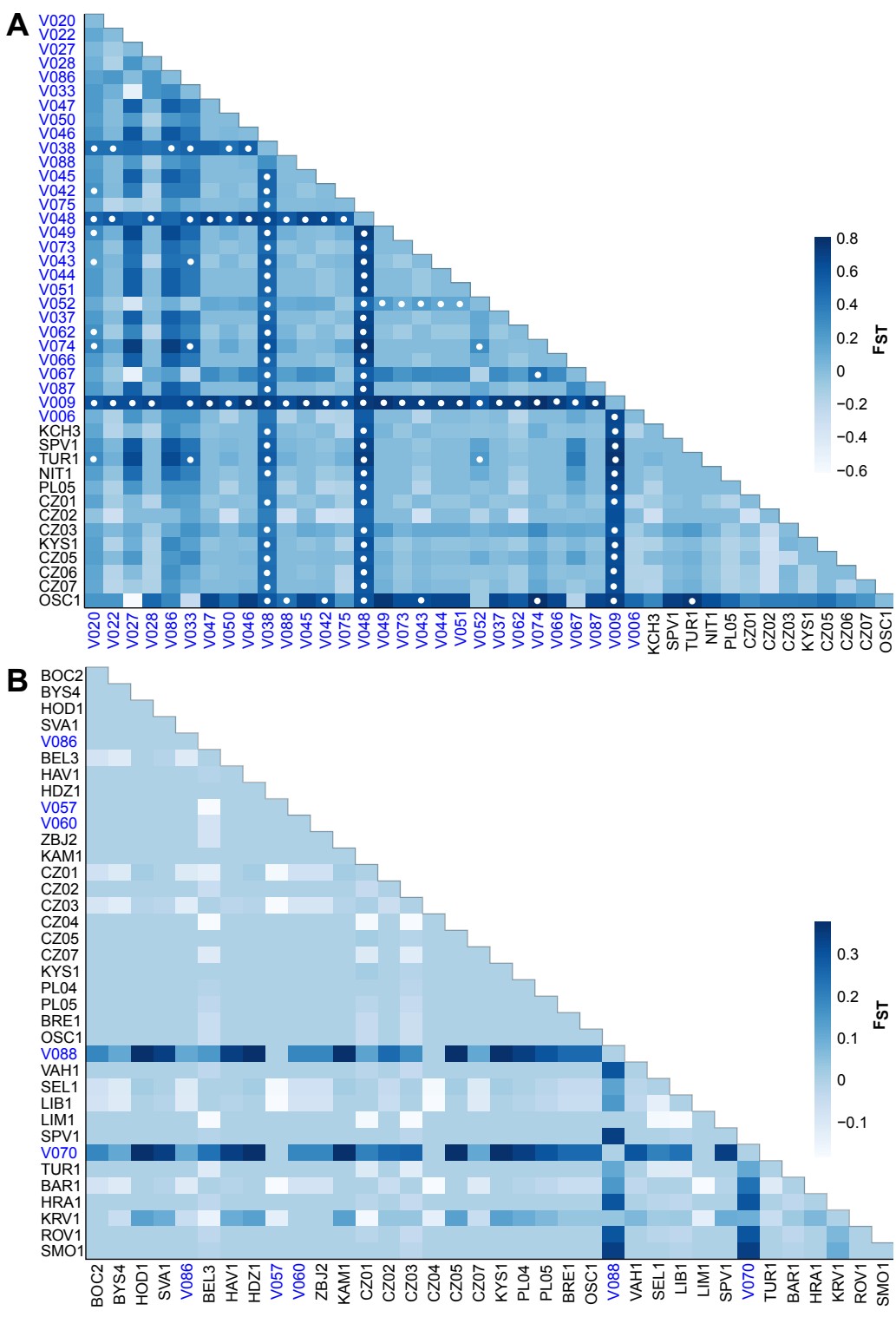

**Figure 5   Heat map of pairwise F$_{ST}$ values among the studied sites (subpopulations) of (A)** *Elmis aenea* **and (B)** *Limnius perrisi* **in the W Carpathians.**  Darker shades of blue rectangles indicate higher values of F$_{ST}$ (as displayed on the bar right of the heat map). White dots indicate F$_{ST}$ $p$-values significantly different from zero ($P < 0.05$). The spring localities are distinguished by the blue color of the font.

**Table 2** Values of neutrality tests (Fu's Fs, Tajima's D, Fu and Li's D test with *p*-values for *Elmis aenea* and *Limnius perrisi* mtDNA COI sequences.

| Species | Fu's $F_s$ test (*p*-value) | Tajima's *D* test (*p*-value) | Fu and Li's *D* test (*p*-value) |
|---|---|---|---|
| *Elmis aenea* | −17.331 (0.000) | −2.047 (0.001) | −3.323 ($< 0.02$) |
| *Limnius perrisi* | −14.064 (0.000) | −2.004 (0.002) | −3.320 ($< 0.02$) |

changes of the aquatic environment, manifested, for example, by the loss of macrophytes and moss (*Maitland, 1967*; *Bradley & Ormerod, 2001*; *Hoffsten, 2003*). These findings may explain much greater affinity of *E. aenea* to springs that generally, with respect to chemical, physical and trophic conditions, are more stable ecosystems compared to other lotic habitats (*Minshall & Winger, 1968*; *Odum, 1971*; *Butler & Hobbs, 1982*; *Cushing & Wolf, 1984*; *Glazier & Gooch, 1987*; *Gooch & Glazier, 1991*). This suggests that karst springs ensured a suitable environment for survival of some aquatic species even during the ice age (*Thorup & Lindegaard, 1977*). It supports the dinodal hypothesis (*Malicky, 1983*; *Malicky, 2000*) proposing that suitable aquatic habitats, persisted throughout the Pleistocene within the periglacial area (dinodal biome), providing suitable conditions for the survival of specialized oligo-stenotherm communities in Central Europe. However, based on our data from the W Carpathians only, the dinodal hypothesis cannot be unequivocally confirmed or refuted, but it clearly opens up new questions in the field of historical-molecular patterns of elmid species in the W Carpathians.

Different results of the Bayesian Skyline Plot analyses between *E. aenea* and *L. perrisi* confirmed an exceptional position of the springs. The springs could have a special status in terms of providing stable environmental conditions irrespective of the climatic changes even during the glacial and interglacial periods which did not provoke a dramatic decline or increase of the *E. aenea* population size in the W Carpathians. In contrast, the populations of *L. perrisi*, occurring predominantly in streams, began to expand rapidly after the LGP. At the beginning of the Holocene (about 11.5–7.5 ka), a thermal maximum was recorded, which probably enhanced the expansion of species (*Dabkowski et al., 2019*), which corresponds to sudden expansion of *L. perrisi*. At that time, local W Carpathian glaciers disappeared completely (*Lindner et al., 2003*), which led to opening of new migration routes and likely also accelerated species dispersal. Early-Holocene warming is thought to be a major driving force for population divergence in temperate species (*Hewitt, 1999*). On the other hand, differences in genetic diversity among species, as recorded between *E. aenea* and *L. perrisi* may be also influenced by variation in diversification rates (*Ricklefs, 2007*; *Stadler, 2011*).

The *E. aenea* population has occurred in springs at a significantly higher rate, corresponding to its higher molecular diversity compared to *L. perrisi* that prefers streams and its population is much more uniform. In line with our results, populations of two cofamilial caddisfly species in south-eastern UK showed contrasting genetic patterns. *Polycentropus flavomaculatus* showed much more pronounced genetic structure than *Plectrocnemia conspersa* in the same region (*Wilcock et al., 2007*). In another study on caddisflies of the Central European highlands, *Drusus discolor* contained three times more haplotypes than *Hydropsyche tenuis*. Such findings suggest that the isolation of *D. discolor*

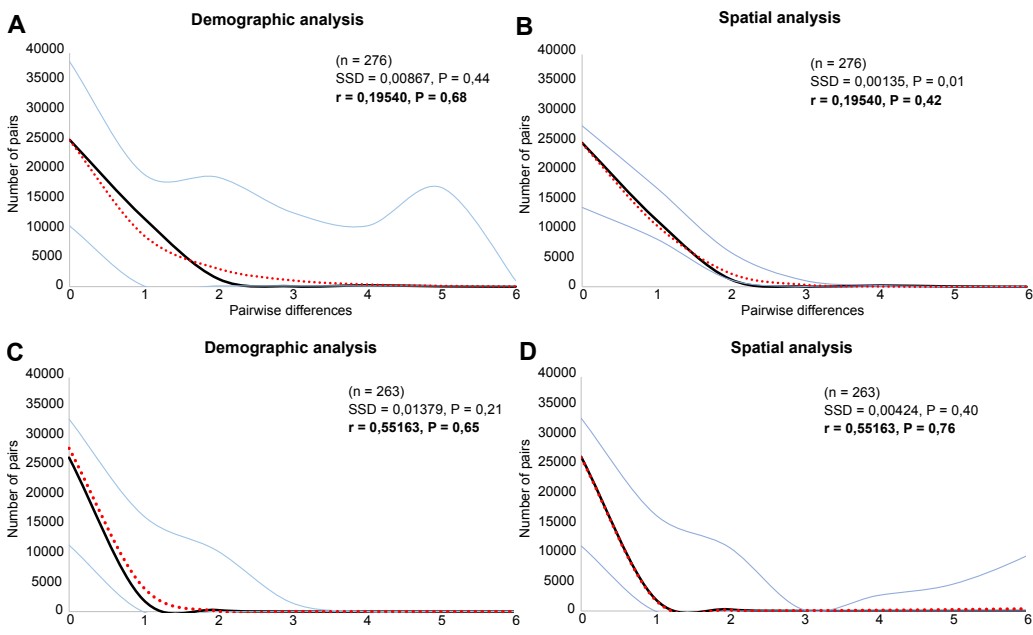

**Figure 6 Mismatch distribution analysis of (A, B) *Elmis aenea* and (C, D) *Limnius perrisi* of the W Carpathian populations based on mtDNA.** Each plot shows the number (*Y* axis) of pairwise nucleotide site differences (*X* axis) among sequences for each species. The fit to the demographic expansion model is evaluated by the SSD and the *r*. The solid black line corresponds to the observed frequency of pairwise differences, the dotted red line represents the pattern expected under a model of sudden demographic expansion. The blue lines are the upper and lower boundaries of the 95% confidence interval.

populations in Central Europe is stronger and persists for a longer time than in *H. tenuis* (*Lehrian, Pauls & Haase, 2009*). Both cases, similarly with species studied herein, confirm that related and co-occurring species may currently have significantly different patterns of molecular diversity, reflecting the different phylogeographical histories of the species and their different autecological traits (*Wilcock et al., 2007*; *Lehrian, Pauls & Haase, 2009*).

Compared to aquatic species occurring in streams, the species preferring springs are generally unable to spread extensively and likely persisted at the foothills of mountains during unfavorable climatic conditions (*Schmitt, 2007*). As a consequence, many of the geomorphological units of the European mountain systems have their own genetic lineages or at least private haplotypes. In our study, we recorded significantly higher values of molecular diversity and higher number of private haplotypes in *E. aenea*. In addition, analysis of spatial autocorrelation for *E. aenea* in springs was significant and consistent with these results. This suggests subpopulations in springs persisted in the study area for a longer time, and are relatively isolated. Conversely, stream subpopulations are more homogeneous and smaller, suggesting that they are probably more recent and are being re-created when environmental conditions improve. Valuable examples documenting the importance of the W Carpathians in terms of biodiversity richness were recent discoveries of local endemism of cold-adapted gammarids from *Gammarus balcanicus* (*Mamos et al., 2014*; *Mamos et al., 2016*) and *Gammarus fossarum* species complexes (*Copilaş-Ciocianu et al., 2017*) or caddisfly species *Drusus discolor*. The latter persisted in the Tatra Mts in

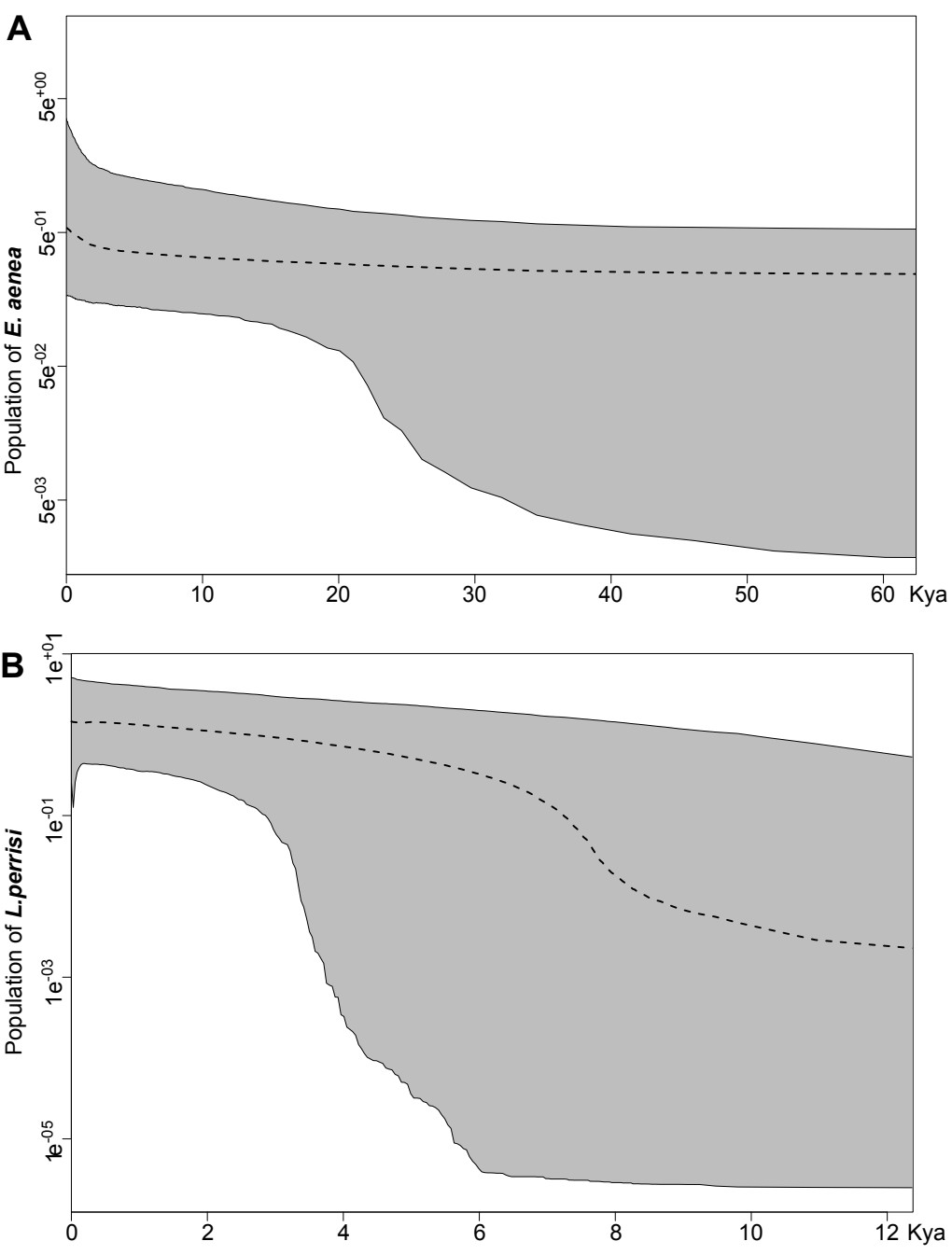

**Figure 7** Extended Bayesian skyline plot based on mtDNA sequences of (A) *Elmis aenea* and (B) *Limnius perrisi* from investigated springs and streams of the W Carpathians, reconstructing the population size history using an evolutionary rate 0.0115 substitutions/site/Myr. The *x*-axis is depicted on a scale of thousands of years (Kya), while *Y*-axis corresponds to the mean effective population size. The dotted line represents the mean, while grey-shaded areas encompass 95% highest posterior density (HPD).

numerous refugia over multiple glacial cycles, allowing many local endemic clades to form (*Pauls, Lumbsch & Haase, 2006*). In the case of *E. aenea*, the two localities in the Slovak Ore Mts (SOM: V038, V048) and one locality in the Fatra-Tatra area (FTA: V009) had remarkably high $F_{ST}$ values, suggesting that some W Carpathian springs could constitute Pleistocene refugia. According to several studies, the role of the W Carpathians as a glacial refugium (*Jamřichová, Potučková & Horsák, 2014*; *Mráz & Ronikier, 2016*; *Jamřichová, Petr & Jiménez-Alfaro, 2017*) for various species or genetic lineages is undoubted (*Pinceel et al., 2005*; *Magri et al., 2006*; *Wielstra, Babik & Arntzen, 2015*; *Mamos et al., 2016*; *Copilaş-Ciocianu et al., 2017*). However, to test whether this hypothesis also applies to Elmidae riffle beetles requires further study in a broader geographical context.

Overall, the genetic differences between populations from different geomorphological subunits of the W Carpathians were very low in both elmid species. However, higher $F_{ST}$ values in *E. aenea* correlated with the results of AMOVA. Genetic differentiation among *E. aenea* subpopulations within geomorphological subunits was relatively high (34.99%). This indicates that there are some well-pronounced differences in genetic composition among most of the spring subpopulations of *E. aenea* within each geographical unit. Similar results emerged from the study on the black fly *Prosimulium neomacropyga* in the US Southern Rockies ecoregion with alpine tundra streams, where the differences among streams within the region were 24.58% (*Finn et al., 2006*). In both elmid species, only a single haplotype was abundant and widespread along the W Carpathians, surrounded by several rare peripheral haplotypes in a star-shaped topology. Similarly, lack of deeper genetic population structure was also found in the W Carpathian populations of the blackfly *Simulium degrangei* (*Jedlička et al., 2012*). The maintenance of intraspecific genetic diversity is generally very important for the adaptation potential and long-term survival of species (*Spielman, Brook & Frankham, 2004*; *Frankham, 2005*). However, prolonged persistence is possible even despite low levels of genetic diversity (*Johnson et al., 2009*). Relatively homogeneous population patterns of both studied riffle beetles may reflect their short history in the W Carpathians. The comparatively low genetic differentiation among populations of trickle midges (Diptera: Thaumaleidae) in Northern Europe was also explained by relatively recent, possibly post-glacial dispersal (*Haubrock et al., 2017*).

## CONCLUSIONS

In conclusion, it seems that different habitat preferences of the two related aquatic beetle species *E. aenea* and *L. perrisi* preserved their similar population-geographical patterns, but shifted their molecular diversity, as well as the time and character of their distribution in the W Carpathians. *E. aenea*, with higher molecular diversity, occurred mainly in the springs compared to the genetically more homogenous population of *L. perrisi* that was found mostly in streams. These findings support the attribution of the W Carpathian springs to potential refugia with a suitable environment allowing for survival of aquatic biota even during the unfavorable climatic conditions through geological ages and maintaining or even developing its intraspecific genetic diversity. In addition, an isolation of the W Carpathians springs is also indicated by the significant results of Mantel and spatial

autocorrelation analysis in *E. aenea*. This study added new information about understudied riffle beetle fauna of one of the world's biodiversity hotspots, the W Carpathians. However, further studies should include more samples from Southern and Eastern Europe in order to understand the holistic biogeographic pattern of the target species and spring fauna in general.

## ACKNOWLEDGEMENTS

We would like to thank Darina Arendt for help with laboratory work, Maroš Kubala for his help in the statistical data processing and the working team of the Department of Ecology, Comenius University in Bratislava, who performed fieldwork with us. Thanks also go to the Erasmus+ program within which Jana Bozáňová carried out part of the research in the laboratory of the Department of Invertebrate Zoology and Hydrobiology of the University of Łódž.

### Funding

This study was supported by the Slovak National Grant Agency VEGA 2/0030/17, VEGA 1/0127/20 and Miniatura 2017/01/X/NZ8/01607 (Polish NCN) as well as by the statutory funds of the University of Łódž. Tomasz Mamos was supported by the Scholarship of the Polish National Agency for Academic Exchange (NAWA) at Bekker Programme (project nb. PN/BEK/2018/1/00225). The funders had no role in study design, data collection and analysis, decision to publish, or preparation of the manuscript.

### Grant Disclosures

The following grant information was disclosed by the authors:
Slovak National Grant Agency:  VEGA 2/0030/17, VEGA 1/0127/20.
Polish NCN: Miniatura 2017/01/X/NZ8/01607.
Scholarship of the Polish National Agency for Academic Exchange (NAWA) at Bekker Programme:  PN/BEK/2018/1/00225.

### Competing Interests

The authors declare there are no competing interests.

### Author Contributions

- Jana Bozáňová conceived and designed the experiments, performed the experiments, analyzed the data, prepared figures and/or tables, authored or reviewed drafts of the paper, and approved the final draft.
- Zuzana Čiamporová Zaťovičová, Fedor Čiampor Jr and Michał Grabowski conceived and designed the experiments, authored or reviewed drafts of the paper, and approved the final draft.
- Tomasz Mamos conceived and designed the experiments, analyzed the data, authored or reviewed drafts of the paper, and approved the final draft.
## Field Study Permissions

The following information was supplied relating to field study approvals (i.e., approving body and any reference numbers):

The sampling was performed in the framework of a broader research, which was permitted on the basis of the permit issued by The District Office, Department of Environmental Care, No: OU-TN-OSZP1-2015/001937-12/Du.

## DNA Deposition

The following information was supplied regarding the deposition of DNA sequences:

The *Elmis aenea* sequences are available at GenBank: HM401299 to HM401301; HM422034; KJ963580; KJ965229; KU906417; KU909631; KU910145; KU910765; KU910942; KU911055; KU911246; KU911807; KU914345; KU914684; KU916340; KU918081; MT357256 to MT357555.

The *Limnius perrisi* sequences are available at GenBank: KU907434; KU908125; KU909092; KU910262; KU911994; KU912101; MT357997 to MT358260.

The FASTA sequence files of both species are also available in the Supplemental Files.

## Data Availability

The *Elmis aenea* dataset named DS-SKELMAEN is available at BOLD systems: DOI: dx.doi.org/10.5883/DS-SKELMAEN.

The *Elmis aenea* sequences are available at GenBank: HM401299 to HM401301; HM422034; KJ963580; KJ965229; KU906417; KU909631; KU910145; KU910765; KU910942; KU911055; KU911246; KU911807; KU914345; KU914684; KU916340; KU918081; MT357256 to MT357555.

The *Limnius perrisi* dataset named DS-SKLIMPER is available at BOLD systems: DOI: dx.doi.org/10.5883/DS-SKLIMPER.

The *Limnius perrisi* sequences are available at GenBank: KU907434; KU908125; KU909092; KU910262; KU911994; KU912101; MT357997 to MT358260.

## Supplemental Information

Supplemental information for this article can be found online at http://dx.doi.org/10.7717/peerj.10039#supplemental-information.

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
