# Peer review of "The tale of springs and streams: how different aquatic ecosystems impacted the mtDNA population structure of two riffle beetles in the Western Carpathians"

_PeerJ, doi:10.7717/peerj.10039_

## Round 0.1 · original submission · Major Revisions

· Academic Editor

Major Revisions

Dear Dr. Bozáňová and colleagues:

Thanks for submitting your manuscript to PeerJ. I have now received three independent reviews of your work, and as you will see, the reviewers raised some concerns about the research. Despite this, these reviewers are optimistic about your work and the potential impact it will have on research studying riffle beetle stream ecology and systematics. Thus, I encourage you to revise your manuscript, accordingly, taking into account all of the concerns raised by all three reviewers.

Importantly, please ensure that an English expert has edited your revised manuscript for content and clarity.

Please also ensure that your figures and tables contain all of the information that is necessary to support your findings and observations. Revise incorrect information. The Materials and Methods appear to be missing important information. All statistical methods should be adequately described such that they are repeatable.

There are many suggestions, which I am sure will greatly improve your manuscript once addressed. I agree with the concerns of the reviewers, and thus feel that their suggestions should be adequately addressed before moving forward.

I look forward to seeing your revision, and thanks again for submitting your work to PeerJ.

Good luck with your revision,

-joe

Reviewer 1 ·

Basic reporting

I have carefully read read the paper entiled „The tale of springs and streams. How different refugial ecosystems impacted the present molecular population structure of two riffle beetles in the Western Carpathians“.
The study is interesting in the aspect of phylogeography of aquatic animals in Europe. Authors used the largest sequence data on the riffle beetles in Europe, and the main hypotheses of the study are testable by performed analyses (excepting question b-Line 102). The study will be a valuable addition to knowledge how different animal taxa survived Pleistocene glaciations, and authors also stressed out the conservation implications of their findings. Writting style is clear and proffesional. Materials and methods are well described. Sampling strategy was well chosen to investigate differences in distribution patterns and genetic structure of two elmid species. Authors also made a comprehensive comparison with other research studing influence of Pleistocene glaciations on aquatic invertebrate taxa.

Here I comment on specific points.
Title
I propose to change „refugial ecosystem“ to something more specific e.g. Pleistocene refugial biotopes/environments/habitats … Springs and streams cannot be considered as separate ecosystems because the flow of matter and energy between them is high. In any case, there is no single definition of what constitutes an ecosystem so it is better to use more specific terms such as habitat or biotope.

Lines 60-63. The sentence is not easy read and understand. Please rewrite this sentence. I suggest „In this context, studies upon the diversity and evolutionary history of aquatic biota studies should be more focused on springs, that are now heavily understudied compared to other aquatic biotopes.“

Lines 88-90. Please add „aquatic beetle family Elmidae“ in this sentence. Maybe to change „of two riffle beetles“ to „two members/species of aquatic beetle family Elmidae“ or „two aquatic beetles of family Elmidae“

Lines 102-103. It is questionable that connectivity between subpopulations can be tested only using COI mtDNA marker. This marker cannot be good for estimation of demographic history over thousands of year but recent connectivity must be analysed with more variable markers such as microsatellites. Better to remove this question. Moreover, I did not found any analyses of gene flow to test connectivity between subpopulations

Line 131. Add „Trenčín (Slovakia)” after “Environmental Care”

Line 251. Please add GenBank Acc. No. of this problematic sample

Figures
>>> Figure 1. It is not clear from presented maps what is the exact position and size of examined area. Also, Switzerland have the same color as seas.

>>> Enlarge fonts in every figure in case that text is hard to read.

>>> Please add scale in kilometres in all figure with geographic maps.

>>> Legends in figures are not well designed. Rectangles in various colors which represent legend for haplotypes have to be separated from squares which represent geomorphological units… It can be explained as text what these rectangles are showing.

>>> Identical colours used in subfigures in Figure 2 and 3 are representing different things: Haplotypes and Geographic areas. Every occurrence of color must be explained by text not only by graphical legend.

>>> Figure 4. Fonts in figure are small. Consider to enlarge them to make them readable, especialy for E. aenea. If enlarged text overlap, it possible to organise text differently, e.g. move laterally every second word or use arrows for indication.

>>> Figure 6. change the time scale from Mya to show value in Kya

Experimental design

No comment

Validity of the findings

No comment

·

Basic reporting

The language can be improved with the help of a native speaker.
The data are insufficient to test the glacial refugia hypothesis. I suggest instead that the manuscript focuses on the ecological and genetic differentiation patterns among the two species in the context of different dispersal capabilities.

Experimental design

no comment

Validity of the findings

The conclusions should reflect the main findings of the study, not discuss additional topics which were not touched anywhere else in the manuscript.

Additional comments

Bozanova and colleagues compared the mitochondrial phylogeographic structure of two Elmidae species that apparently have distinct habitat preferences (one prefers springs, the other streams). The results show that the spring species exhibited a higher genetic differentiation and higher genetic diversity than the stream species. Consequently, the authors propose that springs functioned as refugia during the Pleistocene climatic fluctuations. Overall I find the study interesting, but its premise is flawed. Although the sampling is regionally very good, it is insufficient to test the glacial refugia hypothesis because the sampling outside the W Carpathians is extremely sparse (just a few locations). These two species have very broad geographical ranges. Therefore, with the data at hand it is impossible to reasonably assess the endemicity of the detected haplotypes, and when did they diverged from other haplotypes throughout the ranges of both species. Furthermore, all the analyses unambiguously indicate a rapid and recent postglacial expansion in both species, but it is not clear from where they expanded, the W Carpathians or from more southern areas? This is a critical question, but impossible to test with the current data.

To make the manuscript more coherent and stronger, the part of survival in refugia and adjacent analyses (BSP and time trees) should be discarded and the focus should be more on the comparison of springs vs. streams. In this case the data is stronger, indicating that the spring species has a higher genetic diversity and more geographically structured genetic variation. This is most likely due to different dispersal abilities and ecological tolerance. The geographical coverage is excellent for such a comparison. More focus on ecology (emphasizing more convincingly the habitat preferences of both species) and comparisons of the two species’ genetic differentiation/diversity using formal statistical tests would strengthen the manuscript considerably.

Major points:
1) Although the language is relatively clear, the text could be improved. The use of the definite article is often inappropriate. The flow of the Discussion is rather chaotic, jumping from one idea to another. I suggest that a more experienced/native English speaker should have a look at the text.

2) The formulated questions are rather clear, however, question a) (“are the spring subpopulations
genetically more variable if compared to subpopulations in the streams?”) has not been addressed in the study. Genetic variation of spring and stream populations was not tested. What was tested was genetic variation among two species that seem to have different habitat preferences.

3) The two species are considered ecologically distinct, one preferring springs and the other streams. However, there is no formal analysis which supports this. Some numbers are given in paragraph 204-211. Although a trend is visible, it is not clear whether these differences are significant or not. As such, I suggest that the authors should compare the number of spring and stream localities for each of the two species with a 2x2 Fisher’s exact test. If more ecological data is available, such as water temperature and speed, etc, it would be worthwhile to include it. It would significantly strengthen the study if, for example, it shows that the spring species prefers lower temperatures and slower water speed. Alternatively, if such data is not available, important climatic variables can be extracted for each sampling point using DIVA-GIS from the WordClim database (https://www.worldclim.org/data/index.html). Altitude, temperature and precipitation could show significant differences among species. Of course, this is just a suggestion.

4) For AMOVA, why not test the influence of drainages as well? Especially given that these are aquatic species. Also, it is not clear why AMOVA groups were split to match geological units. Is this known to affect population structure in these organisms? Considering that AMOVA indicates almost no variation among geological subunits, it would be worthwhile testing whether drainages have a bigger influence on population structure.

5) While some analyses appear to show differences between species, there is never a formal statistical comparison. For example, the authors present the number of haplotypes and haplotype diversity per species. Other important indices should also be calculated in DnaSP, such as nucleotide diversity, number of polymorphic sites and mean number of nucleotide differences. However, all these indices should be calculated per population (where more than one individual was sequenced) as this would paint a more accurate picture and would allow a statistical comparison between the two species (using a non-parametric test such as Mann-Whitney). Only then the authors can formally conclude that one species had a higher/lower molecular diversity than the other. Such results could be presented in a table format, or, for a greater effect, a figure with boxplots, for example. The same statistical test should be applied to the Fst values. Was the mean Fst value in E. aenea higher than in L. perrisi?

6) The authors could try to do a pairwise Fst comparison within species only between locations in the same habitat, i.e. only among springs or only among streams. Is the mean Fst among springs higher than among streams? Is it statistically significant? This could be a worthwhile comparison in the context of this study. The sample size seems sufficient.

7) Another analysis that could be done is isolation by distance (IBD). In this case, one would expect a higher IBD in the spring species E. aenea.

8) Regarding the BSP plots it is not clear what data was used. If the authors used all the sequences, including outside the W Carpathians, then the plots of the two species are not comparable in the focal geographical context. This is because the coalescent times will differ among species due to the inclusion of divergent haplotypes outside the focal area. For example, in the case of L. perrisi, one of the two clades (Fig 3) is not found in the W Carpathians. Including it in a BSP analysis produces misleading results if interpreted in the context of the W Carpathians. The same likely applies to the other species.

Also the interpretation of the BSP plots is ambiguous. The authors claim that E. aenea had a stable populations size throughout the LGM, but it has an obvious expansion 3000 years ago, while L. perrisi started expanding much earlier, ca. 8000 years ago. How does this support the claim that the former species survived in the W Carpathians and the latter did not? One major difference between species is the coalescence time, but like explained above, this depends on the haplotypes included in the analysis.

9) The Conclusion section should briefly conclude the main findings of the study. Yet, the authors discuss DNA barcoding in the context of conservation and taxonomy and so on. This appeared from nowhere as it was not touched upon in other parts of the manuscript.


Minor remarks:
The formulation in the title “molecular population structure” is ambiguous. It is not clear what “molecular” refers to; it can be any kind of molecule. Instead, I suggest “phylogeographic structure” or “DNA population structure” or something like that. The idea is to emphasize the use of DNA.

Abstract. “whose molecular diversity and phylogeographic patterns have not yet been explored”. Well, this is not true, there are some phylogeographic studies on spring fauna, but still the area is understudied. Thus, this sentence should be rephrased to emphasize that there is insufficient knowledge. “mtDNA barcoding fragment I” should be “mitochondrial DNA barcoding fragment representing the cytochrome c oxidase subunit I gene”.

Throughout the text both “Carpathian Arc” and “Carpathian Arch” are used. The correct one is “arc”.

Line 121. “Apuşeni” should be “Apuseni”. Please correct throughout the text.

Table 1. How many individuals were analyzed per sampling site? Why samples originating outside the W Carpathians are not included in the table? Please also provide GenBank accession numbers.

Table 2. Could be worthwhile to include herein the values for the molecular diversity indices and the number of analyzed individuals per species.

Figure 1. The inset map showing the wider geographical context is unusual. Greece, Austria and Russia are colored as if they are seas. On this map it is also not indicated where the study area is located.

Figure 4. Do the Fst analyses include only W Carpathian localities? Which values are statistically significant? What correction was applied for multiple testing to avoid type I errors? Maximum values don’t say much. What are the mean values? Are they statistically significant?

Figure 5. Indicate what the colors represent.

Reviewer 3 ·

Basic reporting

In general, the manuscript is clearly worded, well structured, well written, based on a large data set, analyzes are correct and conclusions are well-founded. It is an original primary research with well defined questions. Design, materials and methods are clear and fits with the standards. Methods chapter probvide sufficient details. The results and conclusion are important and are of interest, overall the manuscript is recommended for publication. No serious gaps or errors found; comments are mostly on formal issues than about the content. I am not a native speaker, but as far as I can judge, the English of the manuscript is appropriate and fluent, well formulated, easy to read and understand. Literature cited broadly covers the area and provide a great insight of the backround and a comparision with the results: no problems revealed here. Data depository links and permission have been checked and found accessible and fine.

Specific comments
L96-97: ’but potentially with different microhabitat preferences’: What kind of potential differences in microhabitat preferences are known? It would be interesting and useful to give more detail on this here. In the discussion it is only mentioned but also without details. It may worth to detail or delete completely. And what about mesohabitats (riffle, pools)? If differences in microhabitat preferences occur, that is usually reflected in mesohabitat preferences too, which may explain their different habitat preference for streams and springs.
L98-100: ‘In addition, riffle beetles are considered as a good indicator of water quality and perhaps also of climate change (Elliott, 2008), so our study can also provide valuable data for biodiversity conservation.’: It would be worth considering deleting this sentence. In general, and in my opinion, it interrupts the thread. I would definitely do so, of or move it to discussion.
L110: ‘Carpathian Arch’: In the abstract the authors used ‘Arc’. Both can be fine and correct but should be unified throughout the manuscript.
L115-118: It would be useful to include at least some basic numbers here allowing the readers to have an idea about the number of sites without interrupting reading and without checking the figure caption.
L130: ‘which was permitted on the basis of the permit issued by’ please consider rephrasing this.
L250: L. perrisi should be italics

Table 1: I suggest moving this table to supplementary material, it is not essential to include it to the main part.

Figure 1 (map) It must be done again because of mistakes in the inlet map. Austria and Greece are completely missing, such as Russia at the right top corner and also the Kaliningrad region. Also suggested to indicate the area of the bigger map on the inlet using a black rectangle. It would be more informative if streams and springs can be distinguishable by two colours.

Figure 4: It seems to me that the colour shades are not the same for identical values comparing the two parts. It would be better unified with the same colour coding.

Figures, general: It would be useful to give more self-explaining captions, especially where the figure itself does not easy to understand. This would be of particular help to non-expert readers in understanding the main message if the caption pointed out what to really see. Especially true for Figure 4, 5 and 6.

Experimental design

no comment

Validity of the findings

no comment

Additional comments

Everything has been written above. No more comment

---

## Round 0.2 · Minor Revisions

· Academic Editor

Minor Revisions

Dear Dr. Bozáňová and colleagues:

Thanks for revising your manuscript. The reviewers are very satisfied with your revision (as am I). Great! However, there are a few minor edits to make. Please address these ASAP so we may move towards acceptance of your work.

Best,

-joe

·

Basic reporting

no comment

Experimental design

no comment

Validity of the findings

no comment

Additional comments

I congratulate the authors on the thorough revision. The manuscript has significantly improved both in terms of coherence and analyses. The ecological and genetic differentiation among the two species is much more convincing. I recommend the manuscript for publication.

Not sure, but I believe the following paper might be useful when introducing the W Carpathian springs and their long term persistence (lines 57-64): Franko, O., Šivo, A., Richtáriková, M., Povinec, P.P., 2008. Radiocarbon ages of mineral and thermal waters of Slovakia. Acta Phys. Univ. Comen. 49, 125–132.

Reviewer 3 ·

Basic reporting

The authors have adressed all of my previous points raised in the first round of review. All the required information have been added, all mistakes and typos have been corrected. The text and figures have been improved significantly based on the detailed comments from all the three reviewers. Many parts have been restructured thus readability has increased. English also improved. I have no objection to acceptance of the manuscript in its current state.

Experimental design

no comment

Validity of the findings

no comment

Additional comments

No further comments.

---

## Round 0.3 · accepted · Accept

· Academic Editor

Accept

Dear Dr. Bozáňová and colleagues:

Thanks for revising your manuscript based on the concerns raised by the reviewer. I now believe that your manuscript is suitable for publication. Congratulations! I look forward to seeing this work in print, and I anticipate it being an important resource for groups studying riffle beetle stream ecology and systematics. Thanks again for choosing PeerJ to publish such important work.

Best,

-joe